# Universal ^1^H Spin–Lattice NMR Relaxation Features of Sugar—A Step towards Quality Markers

**DOI:** 10.3390/molecules29112422

**Published:** 2024-05-21

**Authors:** Hafiz Imran Fakhar, Adam Kasparek, Karol Kolodziejski, Leonid Grunin, Mecit Halil Öztop, Muhammad Qasim Hayat, Hussnain A. Janjua, Danuta Kruk

**Affiliations:** 1Medicinal Plants Research Laboratory (MPRL), Department of Agricultural Sciences and Technology, Atta-ur-Rahman School of Applied Biosciences (ASAB), National University of Sciences and Technology (NUST), H-12, Islamabad 44000, Pakistan; hfakhar.phdabs15asab@asab.nust.edu.pk (H.I.F.); m.qasim@asab.nust.edu.pk (M.Q.H.); 2Department of Physics and Biophysics, University of Warmia and Mazury, 10-719 Olsztyn, Poland; karol.kolodziejski@uwm.edu.pl; 3Resonance Systems GmbH, D-73230 Kirchheim unter Teck, Germany; mobilenmr@hotmail.com; 4Department of Food Engineering, Middle East Technical University, Ankara 06800, Turkey; mecit@metu.edu.tr; 5Department of Industrial Biotechnology, Atta-ur-Rahman School of Applied Biosciences, National University of Sciences and Technology, Islamabad 44000, Pakistan; hussnain.janjua@asab.nust.edu.pk

**Keywords:** sugar, NMR Relaxometry, NMR Time, FFC NMR, TD NMR

## Abstract

^1^H fast field-cycling and time-domain nuclear magnetic resonance relaxometry studies have been performed for 15 samples of sugar of different kinds and origins (brown, white, cane, beet sugar). The extensive data set, including results for crystal sugar and sugar/water mixtures, has been thoroughly analyzed, with a focus on identifying relaxation contributions associated with the solid and liquid fractions of the systems and non-exponentiality of the relaxation processes. It has been observed that ^1^H spin–lattice relaxation rates for crystal sugar (solid) vary between 0.45 s^−1^ and 0.59 s^−1^, and the relaxation process shows only small deviations from exponentiality (a quantitative measure of the exponentiality has been provided). The ^1^H spin–lattice relaxation process for sugar/water mixtures has turned out to be bi-exponential, with the relaxation rates varying between about 13 s^−1^–17 s^−1^ (for the faster component) and about 2.1 s^−1^–3.5 s^−1^ (for the slower component), with the ratio between the amplitudes of the relaxation contributions ranging between 2.8 and 4.2. The narrow ranges in which the parameters vary make them a promising marker of the quality and authenticity of sugar.

## 1. Introduction

Table sugar is one of the purest organic food products in the world, with a purity of above 99% regardless of its source. Sugar is extracted from sugar beet (*Beta vulgaris*) or sugarcane (*Saccharum officinarum*) on commercial scales [1,2]. Although cane and beet sugars are chemically similar, they differ in terms of their sensory properties [3] and the presence of metabolites [4], as well as their production methods. One of the primary differences between cane and beet sugar is their geographic origin. Cane sugar is predominantly produced in tropical and subtropical regions, such as Brazil, China, India, Mexico, Pakistan, and Thailand, while beet sugar is produced in cooler regions, such as Europe, North America, and Russia [5,6]. This difference in geographic origin leads to minor differences in their chemical composition, as the plants they are derived from perform different physiological and metabolic processes [7,8]. Sugar is the most common type of sweetener and a multifunctional ingredient, and plays a crucial role in the food industry as it provides sweetness, texture, and other functional properties to a wide range of foods and beverages [1,9,10].

Although both cane and beet sugar are based on sucrose, there can be some differences due to the plants and production processes from which they are obtained. Sugar cane contains a notably higher total polysaccharide content, which is 169 ppm, compared to sugar beet, which contains 77 ppm [11,12]. Furthermore, the process of extracting sugar from sugar beet is more complex (requires specialized equipment) than extracting sugar from sugar cane [1]. Therefore, the cost of production of cane sugar is significantly lower than that of beet sugar, which ultimately leads to unjust competition in the markets [13]. Both beet sugar and cane sugar are almost similar in physical as well as chemical properties. They differ with respect to carbon isotopic ratio and the raffinose and theanderose content. The raffinose content is higher in beet sugar, while theanderose is present in cane sugar only [1,12,14,15,16]. 

The aim of this work is to exploit nuclear magnetic resonance (NMR) relaxometry for the purpose of determining characteristic relaxation features of crystal sugar and sugar/water mixtures that can be used as reference parameters for assessing the authenticity and quality of sugar. NMR relaxometry is a powerful method for providing unique information about the structural and dynamical properties of complex systems on the molecular level. One should stress, at this stage, that while “classical” NMR relaxation experiments are performed at a single (high) magnetic field, fast field-cycling (FFC) NMR relaxometry enables performing relaxation experiments over a very broad range of magnetic fields (resonance frequencies). Consequently, one can probe dynamical processes occurring on different time scales (from milliseconds to nanoseconds), in contrast to “classical” relaxation experiments mostly probing fast molecular motion. This unique advantage of FFC NMR relaxometry has been exploited for systems of different complexity, from liquids, via macromolecules, to tissues [17,18,19,20]. FFC NMR relaxometry also becomes gradually appreciated in food science for quality control and characterization of food products, as outlined in [21]. It has already been used to characterize different kinds of cheese based on water mobility [22], to characterize and determine the origin of different types of honey [23], to determine the relationship between the molecular and macroscopic characteristic of different varieties of oils [24], to determine the molecular network of dextran and whey protein-based composite hydrogels [25,26], and to determine the dynamic and structural properties of Haribo and Vidal jelly [27].

In this work, the potential of ^1^H FFC NMR relaxometry has been combined with ^1^H NMR relaxation experiments in time domain (TD NMR). At this stage, an explanation is needed regarding the terminology used. In fact, the relaxation rates obtained by FFC NMR also stem from analyzing time dependences of ^1^H magnetization. The price for varying the resonance frequency is the limit of the relaxation times accessible in FFC NMR relaxometry; one can hardly probe relaxation processes faster than about 1 ms. The experiments referred to as TD NMR relaxation experiments are performed at a single resonance frequency; however, the accessible range of relaxation times is much broader than in FFC NMR, and one can reveal relaxation components not observed in the FFC experiments. These methods are complementary as a combination of probing relaxation processes occurring on a limited timescale versus frequency (FFC) and relaxation processes over a very broad time scale at a single frequency (TD). The terminology “TD NMR” is related to the name of the equipment.

TD NMR has been used extensively to determine the characteristics and qualities of different kinds of food products. For instance, it has been used to determine the adulteration in honey [28,29], vegetables oils [30], and milk [31]. Moreover, it has been used to detect the presence of microorganisms in milk using magnetic label antibodies [32,33]. In addition, it has been used to determine the oil and fat content in agri-food products [34], solid fat content in dairy products [35], and moisture content in meets [36]. Furthermore, it has been used to classify the different varieties of almond based on origin, harvesting, and farming [37], and has also been used to study the hydration behavior of different types of sugars [38]. The combined approach involving TD NMR and FFC NMR relaxation studies for crystal sugar and sugar/water mixtures is, to our knowledge, the first example of jointly exploiting both methods for the purpose of obtaining complementary information on the relaxation (and hence, dynamical, and structural) features of food products. The studies have been performed on 15 sugar samples of different kind and origin. This comprehensive approach has led to determining characteristic relaxation features of sugar that can serve as a reference (a “relaxation signature”) for evaluating authenticity and quality of sugar.

## 2. Results

### 2.1. TD NMR Experiments

Figure 1 shows the time dependencies of the ^1^H magnetization for crystal (solid) sugar (after averaging over the data collected for three samples of each kind). The magnetization data turned out to be double exponential, and were reproduced in terms of the equation:(1)$$M\left(t\right)=A_{s}\exp\left(-R_{1,s}t\right)+A_{f}\exp\left(-R_{1,f}t\right)+B$$
where $R_{1,s}$ and $R_{1,f}$ denote ^1^H spin–lattice relaxation rates characterizing the fast and slow relaxation contributions, respectively ($R_{1,f}>R_{1,s}$); $A_{s}$ and $A_{f}$ are the corresponding amplitudes of the relaxation contributions; and B is a constant determined by the initial magnetization value: $M\left(0\right)=A_{s}+A_{f}+B$.

The obtained relaxation rates are collected in Table 1. The $R_{1}$ values stem from single-exponential fits.

The ratio between the amplitudes of the two relaxation contributions, $A_{s}/A_{f}$, varies between 6.3 and 19.6, in the last case meaning, in fact, a single-exponential relaxation process. Figure 2 shows the bi-exponential and single-exponential fits for these limiting cases. The bi-exponentiality has also been confirmed by F-test. Analogous fits for the remaining data are shown in Appendix A (Figure A1, Figure A2, Figure A3 and Figure A4).

### 2.2. FFC NMR Data

^1^H relaxation data collected for the sugar/water mixtures using the FFC NMR method also show bi-exponential behavior in the whole frequency range for all samples. The result of the decomposition into the individual contributions (Equation (1)) is shown in Figure 3. In the analysis, the ratio between the amplitudes of the relaxation components has been set as frequency-independent.

The obtained relaxation rates, averaged over the whole frequency range, are collected in Table 2. The relaxation rates can be treated as frequency-independent, to a good approximation. The frequency independence implies that the correlation times are not longer than a few ns. The $R_{1}$ values stem from single-exponential fits.

Although, in this case, the ratio $A_{s}/A_{f}$ varies between 2.8 and 4.2 (that confirms the statement that the relaxation process is bi-exponential in all cases), Table 2 also includes the averaged relaxation rate, $R_{1}$, which resulted from single exponential fits. In Figure 4, the decomposition of the magnetization curves for DEWB is shown as an example.

The decompositions for the other cases are shown in Appendix B (Figure A5, Figure A6, Figure A7, Figure A8, Figure A9, Figure A10, Figure A11, Figure A12, Figure A13, Figure A14, Figure A15, Figure A16, Figure A17 and Figure A18); single-exponential fits are included for comparison.

## 3. Discussion

The analysis presented in Section 2 provides a deep insight into the relaxation properties of crystal as well as hydrated sugar. Beginning with crystal sugar, the relaxation rates obtained from single-exponential fits, $R_{1}$, vary between 0.45 s^−1^ and 0.59 s^−1^. The range is narrow and, therefore, can be treated as a quality control parameter; it can easily be exceeded for counterfeited products as a result of a higher water content, or even very slight paramagnetic admixtures. Following this line, the exponentiality of the relaxation process in crystal sugar has been investigated in detail. For this purpose, the bi-exponential function (Equation (1)) has been used to reproduce the TD data. It has turned out that, in majority of the cases, the ratio $A_{s}/A_{f}$ exceeds 10. Although the factor 10 should not be treated arbitrarily, it is reasonable to consider the relaxation process as single-exponential when the contribution of the second component is below 10%. As one can see from Table 1, only for PTBC and RSWB (2) is the ratio below 10, although in the second case the uncertainty of the ratio, 9.0 ± 1.1, can bring it close to this limit. This leaves us with PTBC (brown, cane sugar from Portugal) with a ratio equal to 6.3 ± 0.8 as the only exception. It is worth keeping this finding in mind. As a result of the low ratio (a relatively small contribution of the slow relaxation component), the $R_{1}$ value for PTBC yields an upper limit of 0.59 s^−1^. The sugar samples can be split into the categories: brown cane sugar (3), white cane sugar (3), white beet sugar (8), and unknown source (1). Although the relaxation rate, $R_{1}$, does not significantly vary, it is of interest to analyze this quantity in connection to the sample categories (Table 3). One can see from the table that although the relaxation rate for brown sugar is somewhat higher than for white sugar, the differences are within the uncertainty range.

The relaxation scenario for hydrated sugar is more complex. The relaxation process is, in all cases, bi-exponential, with the $A_{s}/A_{f}$ ratio ranging from 2.35 for PLWB (3) (white beet sugar from Poland) to 4.06 for ITBC (brown cane sugar from Italy). The slow relaxation component, $R_{1,s}$, that can be attributed to the dynamics of the solid fraction of the mixtures (sugar), ranges between 2.09 s^−1^ for PTWC (white cane sugar from Portugal) to 3.47 s^−1^ for DEBC (brown cane sugar from Germany). The limits of the fast relaxation components, $R_{1,f}$, 13.07 s^−1^ and 17.08 s^−1^, have also been reached for PTWC and DEBC, respectively. As a result of the interplay between the $A_{s}/A_{f}$ ratio and the $R_{1,s}$ and $R_{1,f}$ values, the relaxation rate, $R_{1}$, resulting from single exponential fits, ranges from 3.67 s^−1^ for PTBC (brown cane sugar from Portugal) to 4.96 s^−1^ for DEBC. The average values of these parameters are collected in Table 3. Although the $R_{1,s}$, $R_{1,f},$ and $A_{s}/A_{f}$ for brown sugar are somewhat higher than for white sugar, the differences do not exceed the uncertainties of these values. The relaxation rates obtained from the single-exponential fits for brown and white sugar are the same (4.26 s^−1^), with some differences between white cane sugar and white beet sugar (the differences exceed the uncertainty limits).

In summary, both TD and FFC NMR relaxation studies (the first ones performed for crystal sugar, the second ones for hydrated sugar) show characteristic generic relaxation features that can serve as indicators of the authenticity and quality of sugar.

## 4. Materials and Methods

### 4.1. Origin and Type of Samples

Fifteen sugar samples from 8 countries were collected (Table 4). Eight of the samples were labeled as beet sugar, 6 samples as cane sugar, and the source of 1 sample was unknown. The sample code includes the country of origin (the first two letters, e.g., DE for Germany) and the sample characterization (next two letters, e.g., B for brown sugar and C for cane sugar). The numbers in parentheses (e.g., (1)) are used to indicate different manufactures (in case there is more than one).

Time Domain (TD) NMR experiments were performed for crystal sugar. For fast field-cycling (FFC) NMR experiments sugar/water mixtures including 50% (*w*/*w*) of sugar were prepared.

### 4.2. Experimental Details

NMR studies were performed for the samples listed in Table 4. The studies encompassed time domain (TD NMR) and fast field-cycling (FFC NMR) relaxation experiments. As pointed out above, for TD NMR experiments, crystal sugar was used. The spin–lattice relaxation measurements were performed by means of the saturation recovery sequence using a system operating at 20.34 MHz (^1^H)–(Spin Track, Resonance Systems GmbH, Kirchheim/Teck, Germany). The repetition time (relaxation delay) between scans was set to 10 s, the observation time for the T_1_ relaxation curve corresponding to the maximum value of τ in the sequence 90°-τ–90°–FID was set to 13 s; 4 scans were performed for each of 32 logarithmically distributed points in the magnetization curve (^1^H magnetization versus time). The value of each point was calculated as the mean amplitude of the free induction decay (FID). The temperature was set to 25 °C, with an accuracy of 1 °C.

As far as FFC NMR experiments are concerned, the frequency ranges of 10 kHz to 10 MHz were covered using a SMARtracer NMR Relaxometer (Stelar s.r.l., Mede, Italy). The measurements were performed for hydrated samples (50% wt. of water content). The temperature was also set to 25 °C, controlled with an accuracy of 1 °C. For the measurements in a frequency range of 3.5 MHz to 10 MHz, a non-polarized sequence was used. Below 3.5 MHz the sample was pre-polarized. The measurements were performed for 30 frequency values in the indicated range. The corresponding magnetization curves for each frequency include 32 points, logarithmically distributed.

As the purpose of this work is to reveal relaxation features of sugar in the context of potential applications of NMR relaxation studies for assessing sugar quality and authenticity, a complex sample preparation would reduce the advantages of this method. Therefore, we did not apply any procedures to reduce possible effects of paramagnetic ^17^O on ^1^H relaxation.

## 5. Conclusions

^1^H TD and FFC NMR relaxation studies performed for crystal (solid) sugar and sugar/water mixtures, respectively, have revealed characteristic relaxation features of sugar that can be considered as a reference (markers) of the quality and authenticity of sugar. The ^1^H spin–lattice relaxation rates for crystal sugar vary within a narrow range between 0.45 s^−1^ and 0.59 s^−1^, and can be treated as single-exponential (at the resonance frequency of about 20 MHz). In other words, significant deviations from the single-exponential behavior should be treated as suspicious. As far as sugar/water mixtures are concerned, the relaxation process is bi-exponential, with the relaxation components varying between about 13 s^−1^ to 17 s^−1^ (the faster relaxation process) and about 2.1 s^−1^ to 3.5 s^−1^ (the slower relaxation process), with the ratio between the amplitudes of the relaxation contributions ranging between 2.8 and 4.2. The narrow ranges in which the parameters vary are reflected by their average values (Table 3). They characterize the essential relaxation properties of sugar.

In looking for origin of the small differences in the relaxation features of the individual samples, one should note that although beet and cane sugars are chemically similar (sucrose) and the purity of sugar from both sources can reach 99.9%, sugar always contains a small quantity of impurities (including paramagnetic ions), depending on the source, locality, industry, and even from batch to batch in the same industry. The presence of such impurities might contribute to the observed differences. The presence of molasses in brown sugar is also a possible factor.

## Figures and Tables

**Figure 1 molecules-29-02422-f001:**
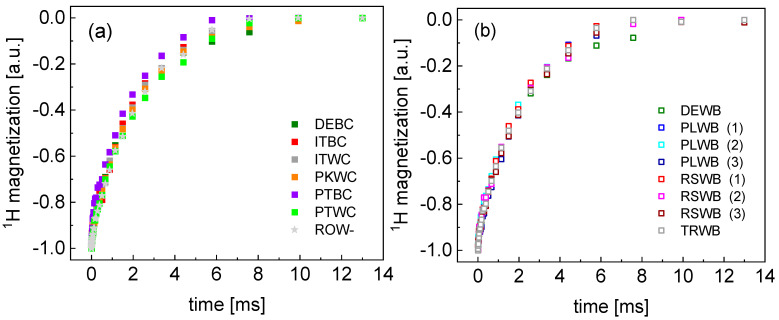
TD ^1^H spin–lattice relaxation data for: (**a**) cane sugar and (**b**) beet sugar. The data for the sample of unknown characteristics are included in (**a**). The codes used are explained in Materials and Methods below.

**Figure 2 molecules-29-02422-f002:**
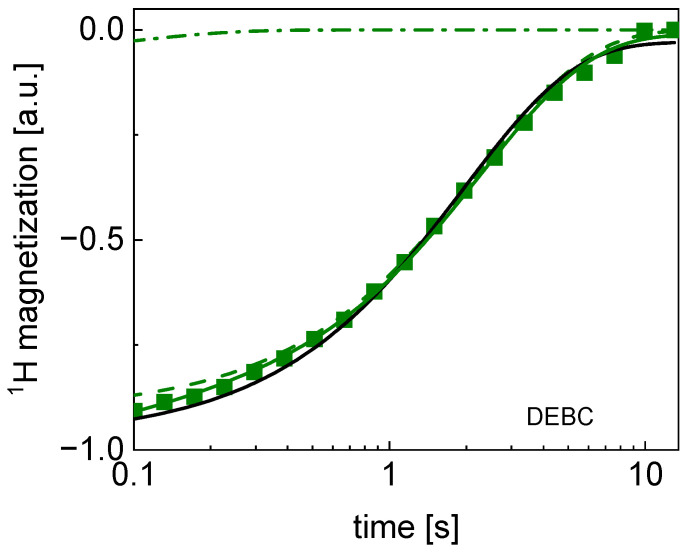
TD ^1^H spin–lattice relaxation data for DEBC sugar decomposed into contribution associated with the $R_{1,s}$ and $R_{1,f}$ relaxation rates. Black line: single exponential fit, solid green line: bi-exponential fit, dashed line: $R_{1,s}$ contribution, dashed-dotted line: $R_{1,f}$ contribution. The uncertainties of the magnetization values are of the order of 5%.

**Figure 3 molecules-29-02422-f003:**
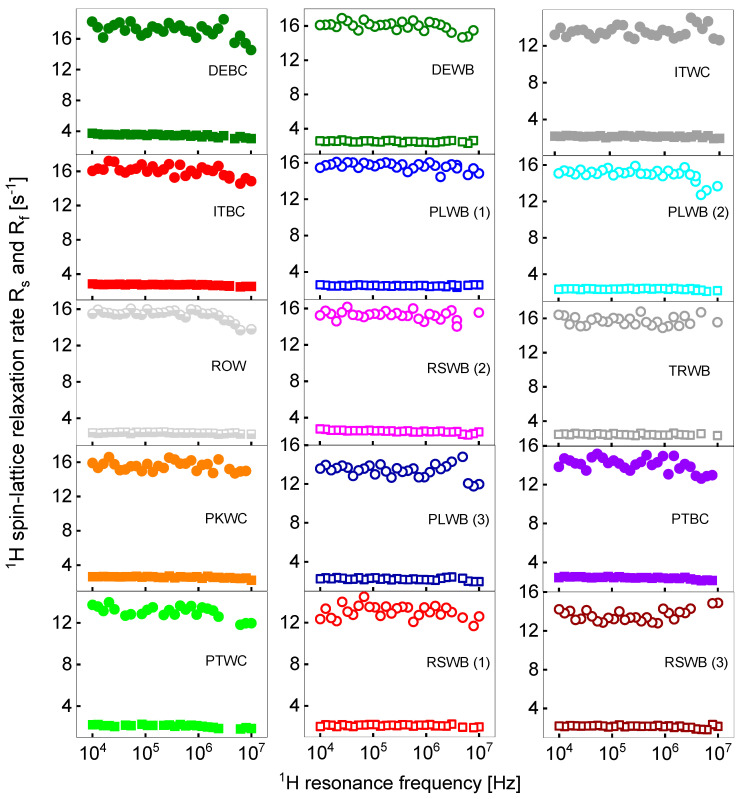
^1^H spin–lattice relaxation data for all sugar sample decomposed into the slow (squares) and the fast (circles) relaxation contributions. The cane sugars are shown in filled squares and circles, the beet sugar in open squares and circles, and the unknown sample is in half-filled squares and circles.

**Figure 4 molecules-29-02422-f004:**
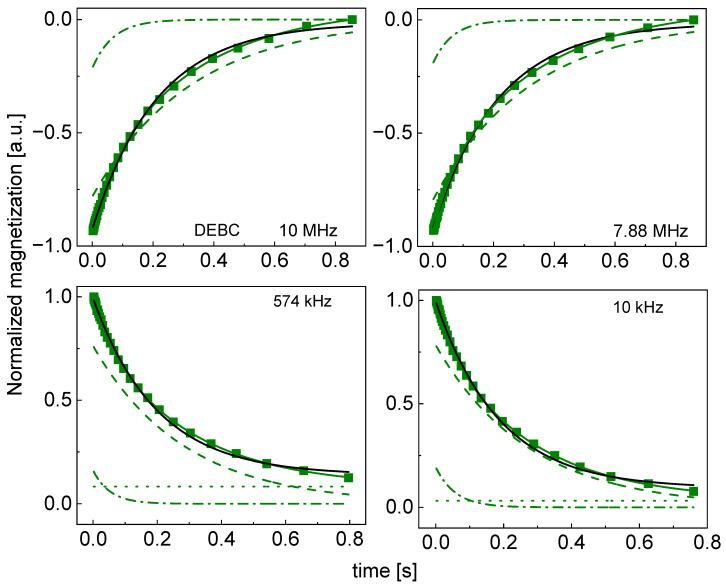
Time evolution of ^1^H magnetization for hydrated DEBC sugar at selected resonance frequencies decomposed into contribution associated with the $R_{1,s}$ and $R_{1,f}$ relaxation rates. Black lines: single-exponential fits, solid green lines: bi-exponential fits, dashed lines: $R_{1,s}$ contributions, dashed-dotted lines: $R_{1,f}$ contributions, dotted lines: B.

**Table 1 molecules-29-02422-t001:** ^1^H spin−lattice relaxation rates for crystal sugar obtained by means of TD NMR.

Sample	$R_{1,s}$ [s^−1^]	$R_{1,f}$ [s^−1^]	$A_{s}/A_{f}$	$R_{1}$ [s^−1^]
DEBC	0.49 ± 0.01	16.9 ± 8.2	13.6 ± 0.9	0.50 ± 0.01
DEWB	0.46 ± 0.01	26.3 ± 2.6	12.9 ± 0.7	0.45 ± 0.02
ITBC	0.43 ± 0.02	34.9 ± 4.6	11.6 ± 2.1	0.50 ± 0.02
ITWC	0.42 ± 0.02	15.7 ± 5.6	13.2 ± 1.4	0.50 ± 0.01
PKWC	0.45 ± 0.01	17.7 ± 2.6	14.6 ± 0.9	0.48 ± 0.03
PLWB (1)	0.39 ± 0.02	18.1 ± 5.8	12.7 ± 1.3	0.49 ± 0.02
PLWB (2)	0.41 ± 0.02	18.9 ± 8.1	13.3 ± 1.6	0.50 ± 0.01
PLWB (3)	0.39 ± 0.01	33.9 ± 9.6	16.1 ± 1.8	0.47 ± 0.02
PTBC	0.44 ± 0.02	22.9 ± 4.7	6.3 ± 0.8	0.59 ± 0.07
PTWC	0.36 ± 0.02	8.60 ± 2.6	10.9 ± 1.2	0.46 ± 0.02
ROW	0.39 ± 0.01	16.6 ± 5.9	19.6 ± 1.7	0.46 ± 0.01
RSWB (1)	0.42 ± 0.01	16.1 ± 5.8	15.2 ± 1.5	0.51 ± 0.01
RSWB (2)	0.39 ± 0.02	14.7 ± 5.0	9.0 ± 1.1	0.51 ± 0.02
RSWB (3)	0.38 ± 0.01	26.5 ± 8.9	12.9 ± 1.4	0.47 ± 0.01
TRWB	0.40 ± 0.01	17.1 ± 4.4	13.2 ± 1.2	0.47 ± 0.02

**Table 2 molecules-29-02422-t002:** Averaged values of ^1^H spin–lattice relaxation rates for sugar/water mixtures obtained by means of FFC NMR.

Sample	$R_{1,s}$ [s^−1^]	$R_{1,f}$ [s^−1^]	$A_{s}/A_{f}$	$R_{1}$ [s^−1^]
DEBC	3.47 ± 0.18	17.08 ± 0.92	3.57 ± 0.60	4.96 ± 0.20
DEWB	2.52 ± 0.09	15.99 ± 0.54	2.81 ± 0.70	4.69 ± 0.11
ITBC	2.74 ± 0.09	16.03 ± 0.66	4.06 ± 0.58	4.16 ± 0.17
ITWC	2.17 ± 0.09	13.52 ± 0.62	2.43 ± 0.75	3.88 ± 0.17
PKWC	2.61 ± 0.10	15.55 ± 0.55	4.15 ± 0.89	4.01 ± 0.17
PLWB (1)	2.49 ± 0.06	15.61 ± 0.42	3.40 ± 0.43	4.33 ± 0.23
PLWB (2)	2.36 ± 0.08	14.96 ± 0.70	2.80 ± 0.73	4.43 ± 0.26
PLWB (3)	2.23 ± 0.12	13.37 ± 0.70	2.35 ± 0.41	4.49 ± 0.24
PTBC	2.42 ± 0.12	14.05 ± 0.70	3.90 ± 0.66	3.67 ± 0.15
PTWC	2.09 ± 0.13	13.07 ± 0.57	2.88 ± 0.20	3.94 ± 0.10
ROW	2.39 ± 0.07	15.37 ± 0.60	2.84 ± 0.37	4.65 ± 0.29
RSWB (1)	2.10 ± 0.09	13.06 ± 0.63	3.15 ± 0.54	3.94 ± 0.06
RSWB (2)	2.49 ± 0.13	15.28 ± 0.47	3.59 ± 0.65	4.17 ± 0.17
RSWB (3)	2.15 ± 0.12	13.61 ± 0.59	2.76 ± 0.73	3.94 ± 0.06
TRWB	2.44 ± 0.08	15.72 ± 0.53	3.15 ± 0.52	4.67 ± 0.11

**Table 3 molecules-29-02422-t003:** Average relaxation parameters for different categories of sugar. Results are averages for each category, with calculated standard error for that category.

Category	Crystal Sugar	Sugar/Water Mixture			
	$R_{1}$ [s^−1^]	$R_{1,s}$ [s^−1^]	$R_{1,f}$ [s^−1^]	$A_{s}/A_{f}$	$R_{1}$ [s^−1^]
brown, cane sugar	0.53 ± 0.03	2.88 ± 0.54	15.7 ± 1.5	3.84 ± 0.25	4.26 ± 0.66
white sugar	0.48 ± 0.02	2.34 ± 0.18	14.6 ± 1.2	3.03 ± 0.50	4.26 ± 0.32
white, cane sugar	0.48 ± 0.02	2.29 ± 0.28	14.0 ± 1.3	3.15 ± 0.89	3.94 ± 0.07
white, beet sugar	0.49 ± 0.02	2.35 ± 0.17	14.7 ± 1.2	3.00 ± 0.40	4.33 ± 0.30

**Table 4 molecules-29-02422-t004:** List of sugar samples with code, origin, and brand label.

Sr. No.	Country	Sample Code	Brand Label	
			Type	Source
1	Germany	DEBC	brown sugar	cane sugar
2		DEWB	white sugar	beet sugar
3	Italy	ITBC	brown sugar	cane sugar
4		ITWC	white sugar	cane sugar
5	Pakistan	PKWC	white sugar	cane sugar
6	Poland	PLWB (1)	white sugar	beet sugar
7		PLWB (2)	white sugar	beet sugar
8		PLWB (3)	white sugar	beet sugar
9	Portugal	PTBC	brown sugar	cane sugar
10		PTWC	white sugar	cane sugar
11	Romania	ROW	white sugar	unknown
12	Serbia	RSWB (1)	white sugar	beet sugar
13		RSWB (2)	white sugar	beet sugar
14		RSWB (3)	white sugar	beet sugar
15	Turkey	TRWB	white sugar	beet sugar

## Data Availability

Data is available in the Zenodo repository: https://doi.org/10.5281/zenodo.11207439.

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
