# Peer review of "Universal 1H Spin–Lattice NMR Relaxation Features of Sugar—A Step towards Quality Markers"

_molecules, 2024, doi:10.3390/molecules29112422_

Round 1

Reviewer 1 Report

Comments and Suggestions for Authors

The authors describe NMR experiments to measure the longitudinal relaxation of various sugar samples. These are differentiated according to the type of source plant (beet or sugar cane), the country of origin and the degree of refining, i.e. brown / white. In addition, the samples were measured in crystallized form and after the addition of water.
Saturation recovery at a proton resonance frequency of 20 MHz is used as the measurement method for granulated sugar and fast field cycling (FFC) between 10 kHz and 10 MHz for mixtures with water. In both cases, the data are fitted monoexponentially as well as biexponentially on a trial basis. The data is then summarized. No further interpretation of the relaxation curves is made. It is claimed that these values could serve as a kind of reference value for testing commercial sugar samples. The authors cite 40 articles, 14 of which they co-authored themselves.
The distinction between the saturation recovery method and FFC with the term "time domain (TD) relaxometry" seems very artificial, because FFC also works in the time domain and is therefore to be assigned to TD relaxometry, except that field switching is used instead of pulses. To describe the results as complementary to each other (line 89) seems somewhat ridiculous, as does "the first example of jointly exploiting both the methods".
The graphical representation of the results also leaves a lot to be desired. For comparisons between monoexponential and biexponential fits, it would be very helpful if error bars were plotted on the data. I suggest displaying the decreasing relaxation curves with a logarithmic ordinate. The biexponential character would be much more visible here. Similarly in Fig. 3: A vertical logarithmic diagram would not make the R1s values appear almost constant, but their relative uncertainties would appear on the same scale as those of R1f.
Further in the caption to Fig. 3: "The unknown sample is in half filled squares and circle" - but there are no such symbols to be seen. This surely refers to the gray symbols at "ROW".
Table 2 shows R1 values obtained using FFC. However, to which frequency do these values belong? Or are these values averaged over all frequencies? There is nothing to be found in the caption or in the relevant subsection 2.2.
Table 3 summarizes all results by averaging over similar samples. The columns are sorted according to the type of experiment ("TD" or FFC). For the reasons given above, this does not appear to make any sense at all. Instead, the sample-related assignment "crystal" - "Water mixture" should appear here.
In a scientific article, it is expected that data is not only collected, but also interpreted. For example, the constancy of the FFC-R1 values between 104 and 106 Hz could indicate extreme narrowing conditions. For some samples, a clear reduction in the R1 values can be observed at higher frequencies. Why not for all samples? Can conclusions be drawn about the correlation times? What could be a reason for the small differences between the samples? What is the possible influence of molasses in brown sugar?
In the present manuscript there is no indication of the possible influence of paramagnetic oxygen on longitudinal relaxation. It is well known that other authors go to some trouble to evacuate their samples or to degas water. The authors should address this and give a reason why they do not consider it necessary to take this into account in the present case.
The authors conclude that these values "show characteristic, generic relaxation features that can serve as indicators of authenticity and quality of sugar" (lines 267, 268). This implies that different values are expected for sugar of lower quality or uncertain origin. However, this assertion can only be substantiated if it is contrasted with measurements on such samples, which is not the case in this article.
Typing errors:
Line 92: ca -> can
Line 237: leaves us
Line 300: Two times: o -> °

Reviewer 2 Report

Comments and Suggestions for Authors

This paper deals with the combined use of 1H time domain (TD) and fast field cycling (FFC) NMR relaxometry analyses applied to samples of sugar (TD) or sugar:water mixtures (FFC), from various countries, types (brown or white) and sources (beet or cane). The analytical method is of interest for readers and deserve to be published.

The abstract and introduction are very well written. Introduction is per se an interesting short review on sugar. It seems to contain all appropriate references needed.

However, I have several remarks on the other parts of the manuscript.

About the mono- or bi-exponential character of magnetization using TD and FFC methods, a rigorous statistical test, for example of a F-test, is needed.

In addition, the way spin-lattice relaxation data are performed is misleading. Table 1 reports R1, the inverse of T1, about 0.5 s-1. This is in high discrepancy with what relate Figures 1, 2 and A1-A4, where the recovery of magnetization stands in the millisecond time scale. The sentence lines 291-293 does not clear this point: "followed by" would mean that the signal of the FID is recorded before a 90°-tau-9° pulse. So, is the FID recorded before or after this module? What are the values of tau? How is this related to the knowledge of longitudinal relaxation? It rather looks like an echo, for the measure of a transverse relaxation rate, more in agreement with the millisecond time scale of the figures. Equation (1) itself is not clear, as it does not resemble a classical expression to extract longitudinal relaxation. One would expect terms of the kind 1-exp(-Rt).

Some results need to be more deeply addressed in the Discussion section.

-          How can one may interpret differences in DT and FFC analysis between samples? In this context, the sentence of lines 263-265 is particularly unclear. From Table 3, the relaxation rates obtained from the single-exponential fits for brown and white sugar are NOT the same; also, "same" should be replaced by "not significantly different". I particularly do not understand the word "same", line 264. Line 265, "the differences" deal about which types of sugar? I hardly find significant differences between types in FFC results, except maybe for As/Af, and here also statistical tests would be useful.

-          Why is there significant discrepancies between samples from the same country, with the same type and source? Are brands different?

In the short conclusion, the authors suggest that the 15 samples of crystal sugar are not suspicious, as mono-exponential relaxation occurs, except for PTBC. Can they have develop arguments about this? Could they complete the analysis with suspicious sugars? The paper indeed does not provide a method significantly distinguishing sugars by their origin, type or source. To strengthen the paper, I would suggest analyzing, by FFC, sugars with different content of water, in order to assessing "authenticity and quality of sugar", the final aim of the study (lines 59-60).

The authors need to reread carefully the manuscript. For example, Table 4 include the type of sugars, either cane or beet. The caption of Figures 1 and 3 indicate systematically the opposite type for each sugar. Moreover, Figure 1a also contain the ROW sample, of unknown origin, and in Figure 3, ROW sample is not in half fill squares and circles. Another example is line 253, as PLWB designs a sugar from Poland, not Serbia.

Here are other minor remarks:

-          Figure 1, please refer to Table 4 for information on sample codes, at it is their first occurrence.

-          Line 119, the minimal As/Af value is 6.3.

-          Figure 4 and B1-B4: "kHz" instead of "KHz".

-          Line 235: "Table 1" instead of "Table 4".

-          Lines 236-240: "PTBC" instead of "PTBS".

-          Line 295: "see above" instead of "see below".

-          Line 304-305: sentence yet mentioned lines 294-295.

-          Lines 309-311: please reformulate the sentence, as analyses have been performed only at about 20 MHz. For example: The 1H spin-lattice relaxation rate at the resonance frequency of about 20 MHz for crystal sugar vary in a narrow range, between 0.45s-1 and 0.59s-1 and can be treated as single-exponential.

Round 2

Reviewer 2 Report

Comments and Suggestions for Authors

The authors adequately answered to my remarks.

However the fact that 1H magnetization on Fig. 2 and A1-A4 varies between -1 and 0, while it varies between 0 and 1 in Fig. 1 may confuse the reader. Probably this comes from the B term in equation 1.

In addition, I would like to signal two typographical errors that make the text unclear:

-          Line 85: "one" instead of "on"

-          Line 287: "some" instead of "same"
